# MOPO: Model-based Offline Policy Optimization

**Tianhe Yu**[*1], **Garrett Thomas**[*1], **Lantao Yu**[1], **Stefano Ermon**[1], **James Zou**[1],
**Sergey Levine**[2], **Chelsea Finn**[†1], **Tengyu Ma**[†1]
Stanford University[1], UC Berkeley[2]
{tianheyu,gwthomas}@cs.stanford.edu

## Abstract

Offline reinforcement learning (RL) refers to the problem of learning policies entirely from a large batch of previously collected data. This problem setting offers the promise of utilizing such datasets to acquire policies without any costly or dangerous active exploration. However, it is also challenging, due to the distributional shift between the offline training data and those states visited by the learned policy. Despite significant recent progress, the most successful prior methods are model-free and constrain the policy to the support of data, precluding generalization to unseen states. In this paper, we first observe that an existing model-based RL algorithm already produces significant gains in the offline setting compared to model-free approaches. However, standard model-based RL methods, designed for the online setting, do not provide an explicit mechanism to avoid the offline setting's distributional shift issue. Instead, we propose to modify the existing model-based RL methods by applying them with rewards artificially penalized by the uncertainty of the dynamics. We theoretically show that the algorithm maximizes a lower bound of the policy's return under the true MDP. We also characterize the trade-off between the gain and risk of leaving the support of the batch data. Our algorithm, Model-based Offline Policy Optimization (MOPO), outperforms standard model-based RL algorithms and prior state-of-the-art model-free offline RL algorithms on existing offline RL benchmarks and two challenging continuous control tasks that require generalizing from data collected for a different task.

## 1 Introduction

Recent advances in machine learning using deep neural networks have shown significant successes in scaling to large realistic datasets, such as ImageNet [13] in computer vision, SQuAD [55] in NLP, and RoboNet [10] in robot learning. Reinforcement learning (RL) methods, in contrast, struggle to scale to many real-world applications, e.g., autonomous driving [74] and healthcare [22], because they rely on costly online trial-and-error. However, pre-recorded datasets in domains like these can be large and diverse. Hence, designing RL algorithms that can learn from those diverse, static datasets would both enable more practical RL training in the real world and lead to more effective generalization.

While off-policy RL algorithms [43, 27, 20] can in principle utilize previously collected datasets, they perform poorly without online data collection. These failures are generally caused by large extrapolation error when the Q-function is evaluated on out-of-distribution actions [19, 36], which can lead to unstable learning and divergence. Offline RL methods propose to mitigate bootstrapped error by constraining the learned policy to the behavior policy induced by the dataset [19, 36, 72, 30, 49, 52, 58]. While these methods achieve reasonable performances in some settings, their learning is limited to behaviors within the data manifold. Specifically, these methods estimate error with respect to out-of-distribution *actions*, but only consider *states* that lie within the offline dataset and do not

---

[*]equal contribution. † equal advising. Orders randomized.

consider those that are out-of-distribution. We argue that it is important for an offline RL algorithm to be equipped with the ability to leave the data support to learn a better policy for two reasons: (1) the provided batch dataset is usually sub-optimal in terms of both the states and actions covered by the dataset, and (2) the target task can be different from the tasks performed in the batch data for various reasons, e.g., because data is not available or hard to collect for the target task. Hence, the central question that this work is trying to answer is: can we develop an offline RL algorithm that generalizes beyond the state and action support of the offline data?

To approach this question, we first hypothesize that model-based RL methods [64, 12, 42, 38, 29, 44] make a natural choice for enabling generalization, for a number of reasons. First, model-based RL algorithms effectively receive more supervision, since the model is trained on every transition, even in sparse-reward settings. Second, they are trained with supervised learning, which provides more stable and less noisy gradients than bootstrapping. Lastly, uncertainty estimation techniques, such as bootstrap ensembles, are well developed for supervised learning methods [40, 35, 60] and are known to perform poorly for value-based RL methods [72]. All

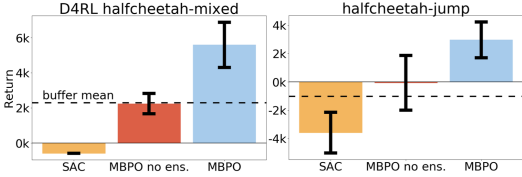

Figure 1: Comparison between vanilla model-based RL (MBPO [29]) with or without model ensembles and vanilla model-free RL (SAC [27]) on two offline RL tasks: one from the D4RL benchmark [18] and one that demands out-of-distribution generalization. We find that MBPO substantially outperforms SAC, providing some evidence that model-based approaches are well-suited for batch RL. For experiment details, see Section 5.

of these attributes have the potential to improve or control generalization. As a proof-of-concept experiment, we evaluate two state-of-the-art off-policy model-based and model-free algorithms, MBPO [29] and SAC [27], in Figure 1. Although neither method is designed for the batch setting, we find that the model-based method and its variant without ensembles show surprisingly large gains. This finding corroborates our hypothesis, suggesting that model-based methods are particularly well-suited for the batch setting, motivating their use in this paper.

Despite these promising preliminary results, we expect significant headroom for improvement. In particular, because offline model-based algorithms cannot improve the dynamics model using additional experience, we expect that such algorithms require careful use of the model in regions outside of the data support. Quantifying the risk imposed by imperfect dynamics and appropriately trading off that risk with the return is a key ingredient towards building a strong offline model-based RL algorithm. To do so, we modify MBPO to incorporate a *reward penalty* based on an estimate of the model error. Crucially, this estimate is model-dependent, and does not necessarily penalize all out-of-distribution states and actions equally, but rather prescribes penalties based on the estimated magnitude of model error. Further, this estimation is done both on *states* and *actions*, allowing generalization to both, in contrast to model-free approaches that only reason about uncertainty with respect to actions.

The primary contribution of this work is an offline model-based RL algorithm that optimizes a policy in an uncertainty-penalized MDP, where the reward function is penalized by an estimate of the model's error. Under this new MDP, we theoretically show that we maximize a lower bound of the return in the true MDP, and find the optimal trade-off between the return and the risk. Based on our analysis, we develop a practical method that estimates model error using the predicted variance of a learned model, uses this uncertainty estimate as a reward penalty, and trains a policy using MBPO in this uncertainty-penalized MDP. We empirically compare this approach, model-based offline policy optimization (MOPO), to both MBPO and existing state-of-the-art model-free offline RL algorithms. Our results suggest that MOPO substantially outperforms these prior methods on the offline RL benchmark D4RL [18] as well as on offline RL problems where the agent must generalize to out-of-distribution states in order to succeed.

## 2 Related Work

Reinforcement learning algorithms are well-known for their ability to acquire behaviors through online trial-and-error in the environment [3, 65]. However, such online data collection can incur high sample complexity [46, 56, 57], limit the power of generalization to unseen random initialization [9, 76, 4], and pose risks in safety-critical settings [68]. These requirements often make real-world applications of RL less feasible. To overcome some of these challenges, we study the batch offline

RL setting [41]. While many off-policy RL algorithms [53, 11, 31, 48, 43, 27, 20, 24, 25] can in principle be applied to a batch offline setting, they perform poorly in practice [19].

**Model-free Offline RL.** Many model-free batch RL methods are designed with two main ingredients: (1) constraining the learned policy to be closer to the behavioral policy either explicitly [19, 36, 72, 30, 49] or implicitly [52, 58], and (2) applying uncertainty quantification techniques, such as ensembles, to stabilize Q-functions [1, 36, 72]. In contrast, our *model-based* method does not rely on constraining the policy to the behavioral distribution, allowing the policy to potentially benefit from taking actions outside of it. Furthermore, we utilize uncertainty quantification to quantify the risk of leaving the behavioral distribution and trade it off with the gains of exploring diverse states.

**Model-based Online RL.** Our approach builds upon the wealth of prior work on model-based online RL methods that model the dynamics by Gaussian processes [12], local linear models [42, 38], neural network function approximators [15, 21, 14], and neural video prediction models [16, 32]. Our work is orthogonal to the choice of model. While prior approaches have used these models to select actions using planning [67, 17, 54, 51, 59, 70], we choose to build upon Dyna-style approaches that optimize for a policy [64, 66, 73, 32, 26, 28, 44], specifically MBPO [29]. See [71] for an empirical evaluation of several model-based RL algorithms. Uncertainty quantification, a key ingredient to our approach, is critical to good performance in model-based RL both theoretically [63, 75, 44] and empirically [12, 7, 50, 39, 8], and in optimal control [62, 2, 34]. Unlike these works, we develop and leverage proper uncertainty estimates that particularly suit the offline setting.

Concurrent work by Kidambi et al. [33] also develops an offline model-based RL algorithm, MOReL. Unlike MOReL, which constructs terminating states based on a hard threshold on uncertainty, MOPO uses a soft reward penalty to incorporate uncertainty. In principle, a potential benefit of a soft penalty is that the policy is allowed to take a few risky actions and then return to the confident area near the behavioral distribution without being terminated. Moreover, while Kidambi et al. [33] compares to model-free approaches, we make the further observation that even a vanilla model-based RL method outperforms model-free ones in the offline setting, opening interesting questions for future investigation. Finally, we evaluate our approach on both standard benchmarks [18] and domains that require out-of-distribution generalization, achieving positive results in both.

## 3 Preliminaries

We consider the standard Markov decision process (MDP) $M = (\mathcal{S}, \mathcal{A}, T, r, \mu_0, \gamma)$, where $\mathcal{S}$ and $\mathcal{A}$ denote the state space and action space respectively, $T(s' \,|\, s, a)$ the transition dynamics, $r(s, a)$ the reward function, $\mu_0$ the initial state distribution, and $\gamma \in (0, 1)$ the discount factor. The goal in RL is to optimize a policy $\pi(a \,|\, s)$ that maximizes the expected discounted return $\eta_M(\pi) := \mathbb{E}_{\pi, T, \mu_0} [\sum_{t=0}^{\infty} \gamma^t r(s_t, a_t)]$. The value function $V_M^\pi(s) := \mathbb{E}_{\pi, T} [\sum_{t=0}^{\infty} \gamma^t r(s_t, a_t) \,|\, s_0 = s]$ gives the expected discounted return under $\pi$ when starting from state $s$.

In the *offline RL* problem, the algorithm only has access to a static dataset $\mathcal{D}_{\text{env}} = \{(s, a, r, s')\}$ collected by one or a mixture of behavior policies $\pi^{\text{B}}$, and cannot interact further with the environment. We refer to the distribution from which $\mathcal{D}_{\text{env}}$ was sampled as the *behavioral distribution*.

We also introduce the following notation for the derivation in Section 4. In the model-based approach we will have a dynamics model $\widehat{T}$ estimated from the transitions in $\mathcal{D}_{\text{env}}$. This *estimated dynamics* defines a *model MDP* $\widehat{M} = (\mathcal{S}, \mathcal{A}, \widehat{T}, r, \mu_0, \gamma)$. Let $\mathbb{P}_{\widehat{T}, t}^\pi(s)$ denote the probability of being in state $s$ at time step $t$ if actions are sampled according to $\pi$ and transitions according to $\widehat{T}$. Let $\rho_{\widehat{T}}^\pi(s, a)$ be the discounted occupancy measure of policy $\pi$ under dynamics $\widehat{T}$: $\rho_{\widehat{T}}^\pi(s, a) := \pi(a \,|\, s) \sum_{t=0}^{\infty} \gamma^t \mathbb{P}_{\widehat{T}, t}^\pi(s)$. Note that $\rho_{\widehat{T}}^\pi$, as defined here, is not a properly normalized probability distribution, as it integrates to $1/(1 - \gamma)$. We will denote (improper) expectations with respect to $\rho_{\widehat{T}}^\pi$ with $\bar{\mathbb{E}}$, as in $\eta_{\widehat{M}}(\pi) = \bar{\mathbb{E}}_{\rho_{\widehat{T}}^\pi}[r(s, a)]$.

We now summarize model-based policy optimization (MBPO) [29], which we build on in this work. MBPO learns a model of the transition distribution $\widehat{T}_\theta(s'|s, a)$ parametrized by $\theta$, via supervised learning on the behavioral data $\mathcal{D}_{\text{env}}$. MBPO also learns a model of the reward function in the same manner. During training, MBPO performs $k$-step rollouts using $\widehat{T}_\theta(s'|s, a)$ starting from state

$s \in \mathcal{D}_{\text{env}}$, adds the generated data to a separate replay buffer $\mathcal{D}_{\text{model}}$, and finally updates the policy $\pi(a|s)$ using data sampled from $\mathcal{D}_{\text{env}} \cup \mathcal{D}_{\text{model}}$. When applied in an online setting, MBPO iteratively collects samples from the environment and uses them to further improve both the model and the policy. In our experiments in Table 1, Table 5.2 and Table 1, we observe that MBPO performs surprisingly well on the offline RL problem compared to model-free methods. In the next section, we derive MOPO, which builds upon MBPO to further improve performance.

# 4 MOPO: Model-Based Offline Policy Optimization

Unlike model-free methods, our goal is to design an offline model-based reinforcement learning algorithm that can take actions that are not strictly within the support of the behavioral distribution. Using a model gives us the potential to do so. However, models will become increasingly inaccurate further from the behavioral distribution, and vanilla model-based policy optimization algorithms may exploit these regions where the model is inaccurate. This concern is especially important in the offline setting, where mistakes in the dynamics will not be corrected with additional data collection.

For the algorithm to perform reliably, it's crucial to balance the return and risk: 1. the potential gain in performance by escaping the behavioral distribution and finding a better policy, and 2. the risk of overfitting to the errors of the dynamics at regions far away from the behavioral distribution. To achieve the optimal balance, we first bound the return from below by the return of a constructed model MDP penalized by the uncertainty of the dynamics (Section 4.1). Then we maximize the conservative estimation of the return by an off-the-shelf reinforcement learning algorithm, which gives MOPO, a generic model-based off-policy algorithm (Section 4.2). We discuss important practical implementation details in Section 4.3.

## 4.1 Quantifying the uncertainty: from the dynamics to the total return

Our key idea is to build a lower bound for the expected return of a policy $\pi$ under the true dynamics and then maximize the lower bound over $\pi$. A natural estimator for the true return $\eta_M(\pi)$ is $\eta_{\widehat{M}}(\pi)$, the return under the estimated dynamics. The error of this estimator depends on, potentially in a complex fashion, the error of $\widehat{M}$, which may compound over time. In this subsection, we characterize how the error of $\widehat{M}$ influences the uncertainty of the total return. We begin by stating a lemma (adapted from [44]) that gives a precise relationship between the performance of a policy under dynamics $T$ and dynamics $\widehat{T}$. (All proofs are given in Appendix B.)

**Lemma 4.1** (Telescoping lemma). *Let $M$ and $\widehat{M}$ be two MDPs with the same reward function $r$, but different dynamics $T$ and $\widehat{T}$ respectively. Let $G_{\widehat{M}}^\pi(s,a) := \mathbb{E}_{s' \sim \widehat{T}(s,a)}[V_M^\pi(s')] - \mathbb{E}_{s' \sim T(s,a)}[V_M^\pi(s')]$.*

*Then,*

$$\eta_{\widehat{M}}(\pi) - \eta_M(\pi) = \gamma \; \bar{\mathbb{E}}_{(s,a) \sim \rho_{\widehat{T}}^\pi} \left[ G_{\widehat{M}}^\pi(s,a) \right] \tag{1}$$

As an immediate corollary, we have

$$\eta_M(\pi) = \bar{\mathbb{E}}_{(s,a) \sim \rho_{\widehat{T}}^\pi} \left[ r(s,a) - \gamma G_{\widehat{M}}^\pi(s,a) \right] \geq \bar{\mathbb{E}}_{(s,a) \sim \rho_{\widehat{T}}^\pi} \left[ r(s,a) - \gamma |G_{\widehat{M}}^\pi(s,a)| \right] \tag{2}$$

Here and throughout the paper, we view $T$ as the real dynamics and $\widehat{T}$ as the learned dynamics. We observe that the quantity $G_{\widehat{M}}^\pi(s,a)$ plays a key role linking the estimation error of the dynamics and the estimation error of the return. By definition, we have that $G_{\widehat{M}}^\pi(s,a)$ measures the difference between $M$ and $\widehat{M}$ under the test function $V^\pi$ — indeed, if $M = \widehat{M}$, then $G_{\widehat{M}}^\pi(s,a) = 0$. By equation (1), it governs the differences between the performances of $\pi$ in the two MDPs. If we could estimate $G_{\widehat{M}}^\pi(s,a)$ or bound it from above, then we could use the RHS of (1) as an upper bound for the estimation error of $\eta_M(\pi)$. Moreover, equation (2) suggests that a policy that obtains high reward in the estimated MDP while also minimizing $G_{\widehat{M}}^\pi$ will obtain high reward in the real MDP.

However, computing $G_{\widehat{M}}^\pi$ remains elusive because it depends on the unknown function $V_M^\pi$. Leveraging properties of $V_M^\pi$, we will replace $G_{\widehat{M}}^\pi$ by an upper bound that depends solely on the error of the

dynamics $\widehat{T}$. We first note that if $\mathcal{F}$ is a set of functions mapping $\mathcal{S}$ to $\mathbb{R}$ that contains $V_M^\pi$, then,

$$|G_{\widehat{M}}^\pi(s,a)| \le \sup_{f \in \mathcal{F}} \left| \mathop{\mathbb{E}}_{s' \sim \widehat{T}(s,a)}[f(s')] - \mathop{\mathbb{E}}_{s' \sim T(s,a)}[f(s')] \right| =: d_\mathcal{F}(\widehat{T}(s,a), T(s,a)), \qquad (3)$$

where $d_\mathcal{F}$ is the integral probability metric (IPM) [47] defined by $\mathcal{F}$. IPMs are quite general and contain several other distance measures as special cases [61]. Depending on what we are willing to assume about $V_M^\pi$, there are multiple options to bound $G_{\widehat{M}}^\pi$ by some notion of error of $\widehat{T}$, discussed in greater detail in Appendix A:

(i) If $\mathcal{F} = \{f : \|f\|_\infty \le 1\}$, then $d_\mathcal{F}$ is the *total variation distance*. Thus, if we assume that the reward function is bounded such that $\forall(s,a),\ |r(s,a)| \le r_{\max}$, we have $\|V^\pi\|_\infty \le \sum_{t=0}^\infty \gamma^t r_{\max} = \frac{r_{\max}}{1-\gamma}$, and hence

$$|G_{\widehat{M}}^\pi(s,a)| \le \frac{r_{\max}}{1-\gamma} D_{\mathrm{TV}}(\widehat{T}(s,a), T(s,a)) \qquad (4)$$

(ii) If $\mathcal{F}$ is the set of 1-Lipschitz function w.r.t. to some distance metric, then $d_\mathcal{F}$ is the *1-Wasserstein distance* w.r.t. the same metric. Thus, if we assume that $V_M^\pi$ is $L_v$-Lipschitz with respect to a norm $\|\cdot\|$, it follows that

$$|G_{\widehat{M}}^\pi(s,a)| \le L_v W_1(\widehat{T}(s,a), T(s,a)) \qquad (5)$$

Note that when $\widehat{T}$ and $T$ are both deterministic, then $W_1(\widehat{T}(s,a), T(s,a)) = \|\widehat{T}(s,a) - T(s,a)\|$ (here $T(s,a)$ denotes the deterministic output of the model $T$).

Approach (ii) has the advantage that it incorporates the geometry of the state space, but at the cost of an additional assumption which is generally impossible to verify in our setting. The assumption in (i), on the other hand, is extremely mild and typically holds in practice. Therefore we will prefer (i) unless we have some prior knowledge about the MDP. We summarize the assumptions and the inequalities in the options above as follows.

**Assumption 4.2.** Assume a scalar $c$ and a function class $\mathcal{F}$ such that $V_M^\pi \in c\mathcal{F}$ for all $\pi$.

As a direct corollary of Assumption 4.2 and equation (3), we have

$$|G_{\widehat{M}}^\pi(s,a)| \le c d_\mathcal{F}(\widehat{T}(s,a), T(s,a)). \qquad (6)$$

Concretely, option (i) above corresponds to $c = r_{\max}/(1-\gamma)$ and $\mathcal{F} = \{f : \|f\|_\infty \le 1\}$, and option (ii) corresponds to $c = L_v$ and $\mathcal{F} = \{f : f \text{ is 1-Lipschitz}\}$. We will analyze our framework under the assumption that we have access to an oracle uncertainty quantification module that provides an upper bound on the error of the model. In our implementation, we will estimate the error of the dynamics by heuristics (see sections 4.3 and D).

**Assumption 4.3.** Let $\mathcal{F}$ be the function class in Assumption 4.2. We say $u : \mathcal{S} \times \mathcal{A} \to \mathbb{R}$ is an admissible error estimator for $\widehat{T}$ if $d_\mathcal{F}(\widehat{T}(s,a), T(s,a)) \le u(s,a)$ for all $s \in \mathcal{S}, a \in \mathcal{A}$.[2]

Given an admissible error estimator, we define the *uncertainty-penalized reward* $\tilde{r}(s,a) := r(s,a) - \lambda u(s,a)$ where $\lambda := \gamma c$, and the *uncertainty-penalized MDP* $\widetilde{M} = (\mathcal{S}, \mathcal{A}, \widehat{T}, \tilde{r}, \mu_0, \gamma)$. We observe that $\widetilde{M}$ is conservative in that the return under it bounds from below the true return:

$$\eta_M(\pi) \ge \mathop{\bar{\mathbb{E}}}_{(s,a)\sim\rho_{\widehat{T}}^\pi} \left[ r(s,a) - \gamma|G_{\widehat{M}}^\pi(s,a)| \right] \ge \mathop{\bar{\mathbb{E}}}_{(s,a)\sim\rho_{\widehat{T}}^\pi} [r(s,a) - \lambda u(s,a)]$$

$$\text{(by equation (2) and (6))}$$

$$\ge \mathop{\bar{\mathbb{E}}}_{(s,a)\sim\rho_{\widehat{T}}^\pi} [\tilde{r}(s,a)] = \eta_{\widetilde{M}}(\pi) \qquad (7)$$

## 4.2 Policy optimization on uncertainty-penalized MDPs

Motivated by (7), we optimize the policy on the uncertainty-penalized MDP $\widetilde{M}$ in Algorithm 1.

**Algorithm 1** Framework for Model-based Offline Policy Optimization (MOPO) with Reward Penalty
---
**Require:** Dynamics model $\widehat{T}$ with admissible error estimator $u(s, a)$; constant $\lambda$.
  1: Define $\tilde{r}(s, a) = r(s, a) - \lambda u(s, a)$. Let $\widetilde{M}$ be the MDP with dynamics $\widehat{T}$ and reward $\tilde{r}$.
  2: Run any RL algorithm on $\widetilde{M}$ until convergence to obtain $\hat{\pi} = \arg\max_\pi \eta_{\widetilde{M}}(\pi)$
---

**Theoretical Guarantees for MOPO.** We will theoretical analyze the algorithm by establishing the optimality of the learned policy $\hat{\pi}$ among a family of policies. Let $\pi^\star$ be the optimal policy on $M$ and $\pi^{\mathrm{B}}$ be the policy that generates the batch data. Define $\epsilon_u(\pi)$ as

$$\epsilon_u(\pi) := \mathop{\bar{\mathbb{E}}}_{(s,a)\sim\rho_{\widehat{T}}^\pi} [u(s, a)] \tag{8}$$

Note that $\epsilon_u$ depends on $\widehat{T}$, but we omit this dependence in the notation for simplicity. We observe that $\epsilon_u(\pi)$ characterizes how erroneous the model is along trajectories induced by $\pi$. For example, consider the extreme case when $\pi = \pi^{\mathrm{B}}$. Because $\widehat{T}$ is learned on the data generated from $\pi^{\mathrm{B}}$, we expect $\widehat{T}$ to be relatively accurate for those $(s, a) \sim \rho_{\widehat{T}}^{\pi^{\mathrm{B}}}$, and thus $u(s, a)$ tends to be small. Thus, we expect $\epsilon_u(\pi^{\mathrm{B}})$ to be quite small. On the other end of the spectrum, when $\pi$ often visits states out of the batch data distribution in the real MDP, namely $\rho_T^\pi$ is different from $\rho_T^{\pi^{\mathrm{B}}}$, we expect that $\rho_{\widehat{T}}^\pi$ is even more different from the batch data and therefore the error estimates $u(s, a)$ for those $(s, a) \sim \rho_{\widehat{T}}^\pi$ tend to be large. As a consequence, we have that $\epsilon_u(\pi)$ will be large.

For $\delta \geq \delta_{\min} := \min_\pi \epsilon_u(\pi)$, let $\pi^\delta$ be the best policy among those incurring model error at most $\delta$:

$$\pi^\delta := \mathop{\arg\max}_{\pi : \epsilon_u(\pi) \leq \delta} \eta_M(\pi) \tag{9}$$

The main theorem provides a performance guarantee on the policy $\hat{\pi}$ produced by MOPO.

**Theorem 4.4.** *Under Assumption 4.2 and 4.3, the learned policy $\hat{\pi}$ in MOPO (Algorithm 1) satisfies*

$$\eta_M(\hat{\pi}) \geq \sup_\pi \{\eta_M(\pi) - 2\lambda\epsilon_u(\pi)\} \tag{10}$$

*In particular, for all $\delta \geq \delta_{\min}$,*

$$\eta_M(\hat{\pi}) \geq \eta_M(\pi^\delta) - 2\lambda\delta \tag{11}$$

**Interpretation:** One consequence of (10) is that $\eta_M(\hat{\pi}) \geq \eta_M(\pi^{\mathrm{B}}) - 2\lambda\epsilon_u(\pi^{\mathrm{B}})$. This suggests that $\hat{\pi}$ should perform at least as well as the behavior policy $\pi^{\mathrm{B}}$, because, as argued before, $\epsilon_u(\pi^{\mathrm{B}})$ is expected to be small.

Equation (11) tells us that the learned policy $\hat{\pi}$ can be as good as any policy $\pi$ with $\epsilon_u(\pi) \leq \delta$, or in other words, any policy that visits states with sufficiently small uncertainty as measured by $u(s, a)$. A special case of note is when $\delta = \epsilon_u(\pi^\star)$, we have $\eta_M(\hat{\pi}) \geq \eta_M(\pi^\star) - 2\lambda\epsilon_u(\pi^\star)$, which suggests that the suboptimality gap between the learned policy $\hat{\pi}$ and the optimal policy $\pi^\star$ depends on the error $\epsilon_u(\pi^\star)$. The closer $\rho_{\widehat{T}}^{\pi^\star}$ is to the batch data, the more likely the uncertainty $u(s, a)$ will be smaller on those points $(s, a) \sim \rho_{\widehat{T}}^{\pi^\star}$. On the other hand, the smaller the uncertainty error of the dynamics is, the smaller $\epsilon_u(\pi^\star)$ is. In the extreme case when $u(s, a) = 0$ (perfect dynamics and uncertainty quantification), we recover the optimal policy $\pi^\star$.

Second, by varying the choice of $\delta$ to maximize the RHS of Equation (11), we trade off the risk and the return. As $\delta$ increases, the return $\eta_M(\pi^\delta)$ increases also, since $\pi^\delta$ can be selected from a larger set of policies. However, the risk factor $2\lambda\delta$ increases also. The optimal choice of $\delta$ is achieved when the risk balances the gain from exploring policies far from the behavioral distribution. The exact optimal choice of $\delta$ may depend on the particular problem. We note $\delta$ is only used in the analysis, and our algorithm *automatically achieves the optimal balance* because Equation (11) holds for any $\delta$.

### 4.3 Practical implementation

Now we describe a practical implementation of MOPO motivated by the analysis above. The method is summarized in Algorithm 2 in Appendix C, and largely follows MBPO with a few key exceptions.

Following MBPO, we model the dynamics using a neural network that outputs a Gaussian distribution over the next state and reward[3]: $\widehat{T}_{\theta,\phi}(s_{t+1}, r | s_t, a_t) = \mathcal{N}(\mu_\theta(s_t, a_t), \Sigma_\phi(s_t, a_t))$. We learn an ensemble of $N$ dynamics models $\{\widehat{T}^i_{\theta,\phi} = \mathcal{N}(\mu^i_\theta, \Sigma^i_\phi)\}_{i=1}^N$, with each model trained independently via maximum likelihood.

The most important distinction from MBPO is that we use uncertainty quantification following the analysis above. We aim to design the uncertainty estimator that captures both the epistemic and aleatoric uncertainty of the true dynamics. Bootstrap ensembles have been shown to give a consistent estimate of the population mean in theory [5] and empirically perform well in model-based RL [7]. Meanwhile, the learned variance of a Gaussian probabilistic model can theoretically recover the true aleatoric uncertainty when the model is well-specified. To leverage both, we design our error estimator $u(s, a) = \max_{i=1}^N \|\Sigma^i_\phi(s, a)\|_F$, the maximum standard deviation of the learned models in the ensemble. We use the maximum of the ensemble elements rather than the mean to be more conservative and robust. While this estimator lacks theoretical guarantees, we find that it is sufficiently accurate to achieve good performance in practice.[4] Hence the practical uncertainty-penalized reward of MOPO is computed as $\tilde{r}(s, a) = \hat{r}(s, a) - \lambda \max_{i=1,...,N} \|\Sigma^i_\phi(s, a)\|_F$ where $\hat{r}$ is the mean of the predicted reward output by $\widehat{T}$.

We treat the penalty coefficient $\lambda$ as a user-chosen hyperparameter. Since we do not have a true admissible error estimator, the value of $\lambda$ prescribed by the theory may not be an optimal choice in practice; it should be larger if our heuristic $u(s, a)$ underestimates the true error and smaller if $u$ substantially overestimates the true error.

## 5 Experiments

In our experiments, we aim to study the follow questions: (1) How does MOPO perform on standard offline RL benchmarks in comparison to prior state-of-the-art approaches? (2) Can MOPO solve tasks that require generalization to out-of-distribution behaviors? (3) How does each component in MOPO affect performance?

Question (2) is particularly relevant for scenarios in which we have logged interactions with the environment but want to use those data to optimize a policy for a different reward function. To study (2) and challenge methods further, we construct two additional continuous control tasks that demand out-of-distribution generalization, as described in Section 5.2. To answer question (3), we conduct a complete ablation study to analyze the effect of each module in MOPO in Appendix D. For more details on the experimental set-up and hyperparameters, see Appendix G. For more details on the experimental set-up and hyperparameters, see Appendix G. The code is available online[5].

We compare against several baselines, including the current state-of-the-art model-free offline RL algorithms. Bootstrapping error accumulation reduction (BEAR) aims to constrain the policy's actions to lie in the support of the behavioral distribution [36]. This is implemented as a constraint on the average MMD [23] between $\pi(\cdot | s)$ and a generative model that approximates $\pi^B(\cdot | s)$. Behavior-regularized actor critic (BRAC) is a family of algorithms that operate by penalizing the value function by some measure of discrepancy (KL divergence or MMD) between $\pi(\cdot | s)$ and $\pi^B(\cdot | s)$ [72]. BRAC-v uses this penalty both when updating the critic and when updating the actor, while BRAC-p uses this penalty only when updating the actor and does not explicitly penalize the critic.

### 5.1 Evaluation on the D4RL benchmark

To answer question (1), we evaluate our method on a large subset of datasets in the D4RL benchmark [18] based on the MuJoCo simulator [69], including three environments (halfcheetah, hopper, and walker2d) and four dataset types (random, medium, mixed, medium-expert), yielding a total of

| Dataset type | Environment | BC | MOPO (ours) | MBPO | SAC | BEAR | BRAC-v |
|---|---|---|---|---|---|---|---|
| random | halfcheetah | 2.1 | **35.4** $\pm$ 2.5 | 30.7 $\pm$ 3.9 | 30.5 | 25.5 | 28.1 |
| random | hopper | 1.6 | 11.7 $\pm$ 0.4 | 4.5 $\pm$ 6.0 | 11.3 | 9.5 | **12.0** |
| random | walker2d | 9.8 | **13.6** $\pm$ 2.6 | 8.6 $\pm$ 8.1 | 4.1 | 6.7 | 0.5 |
| medium | halfcheetah | 36.1 | 42.3 $\pm$ 1.6 | 28.3 $\pm$ 22.7 | -4.3 | 38.6 | **45.5** |
| medium | hopper | 29.0 | 28.0 $\pm$ 12.4 | 4.9 $\pm$ 3.3 | 0.8 | **47.6** | 32.3 |
| medium | walker2d | 6.6 | 17.8 $\pm$ 19.3 | 12.7 $\pm$ 7.6 | 0.9 | 33.2 | **81.3** |
| mixed | halfcheetah | 38.4 | **53.1** $\pm$ 2.0 | 47.3 $\pm$ 12.6 | -2.4 | 36.2 | 45.9 |
| mixed | hopper | 11.8 | **67.5** $\pm$ 24.7 | 49.8 $\pm$ 30.4 | 1.9 | 10.8 | 0.9 |
| mixed | walker2d | 11.3 | **39.0** $\pm$ 9.6 | 22.2 $\pm$ 12.7 | 3.5 | 25.3 | 0.8 |
| med-expert | halfcheetah | 35.8 | **63.3** $\pm$38.0 | 9.7 $\pm$ 9.5 | 1.8 | 51.7 | 45.3 |
| med-expert | hopper | 111.9 | 23.7 $\pm$ 6.0 | **56.0** $\pm$ 34.5 | 1.6 | 4.0 | 0.8 |
| med-expert | walker2d | 6.4 | 44.6 $\pm$ 12.9 | 7.6 $\pm$ 3.7 | -0.1 | 26.0 | **66.6** |

Table 1: Results for D4RL datasets. Each number is the normalized score proposed in [18] of the policy at the last iteration of training, averaged over 6 random seeds, $\pm$ standard deviation. The scores are undiscounted average returns normalized to roughly lie between 0 and 100, where a score of 0 corresponds to a random policy, and 100 corresponds to an expert. We include the performance of behavior cloning (**BC**) from the batch data for comparison. Numbers for model-free methods taken from [18], which does not report standard deviation. We omit BRAC-p in this table for space because BRAC-v obtains higher performance in 10 of these 12 tasks and is only slightly weaker on the other two. We bold the highest mean.

12 problem settings. We also perform empirical evaluations on non-MuJoCo environments in Appendix F. The datasets in this benchmark have been generated as follows: **random**: roll out a randomly initialized policy for 1M steps. **medium**: partially train a policy using SAC, then roll it out for 1M steps. **mixed**: train a policy using SAC until a certain (environment-specific) performance threshold is reached, and take the replay buffer as the batch. **medium-expert**: combine 1M samples of rollouts from a fully-trained policy with another 1M samples of rollouts from a partially trained policy or a random policy.

Results are given in Table 1. MOPO is the strongest by a significant margin on all the mixed datasets and most of the medium-expert datasets, while also achieving strong performance on all of the random datasets. MOPO performs less well on the medium datasets. We hypothesize that the lack of action diversity in the medium datasets make it more difficult to learn a model that generalizes well. Fortunately, this setting is one in which model-free methods can perform well, suggesting that model-based and model-free approaches are able to perform well in complementary settings.

## 5.2 Evaluation on tasks requiring out-of-distribution generalization

To answer question (2), we construct two environments `halfcheetah-jump` and `ant-angle` where the agent must solve a task that is different from the purpose of the behavioral policy. The trajectories of the batch data in the these datasets are from policies trained for the original dynamics and reward functions `HalfCheetah` and `Ant` in OpenAI Gym [6] which incentivize the cheetach and ant to move forward as fast as possible. Note that for `HalfCheetah`, we set the maximum velocity to be 3. Concretely, we train SAC for 1M steps and use the entire training replay buffer as the trajectories for the batch data. Then, we assign these trajectories with new rewards that incentivize the cheetach to jump and the ant to run towards the top right corner with a 30 degree angle. Thus, to achieve good performance for the new reward functions, the policy need to leave the observational distribution, as visualized in Figure 2. We include the exact forms of the new reward functions in Appendix G. In these environments, learning the correct behaviors requires leaving the support of the data distribution; optimizing solely within the data manifold will lead to sub-optimal policies.

In Table 2, we show that MOPO significantly outperforms the state-of-the-art model-free approaches. In particular, model-free offline RL cannot outperform the best trajectory in the batch dataset, whereas MOPO exceeds the batch max by a significant margin. This validates that MOPO is able to generalize to out-of-distribution behaviors while existing model-free methods are unable to solve those challenges. Note that vanilla MBPO performs much better than SAC in the two environments, consolidating our claim that vanilla model-based methods can attain better results than model-free methods in the offline setting, especially where generalization to out-of-distribution is needed. The visualization in Figure 2 suggests indeed the policy learned MOPO can effectively solve the tasks by reaching to states unseen in the batch data. Furthermore, we test the limit of the generalization abilities of MOPO in these environments and the results are included in Appendix E.

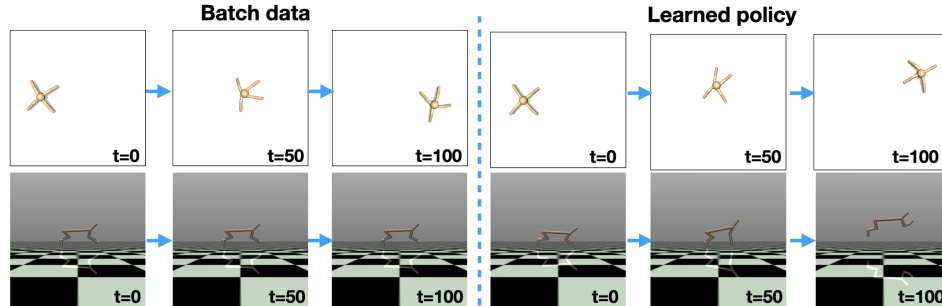

Figure 2: We visualize the two out-of-distribution generalization environments `halfcheetah-jump` (**bottom row**) and `ant-angle` (**top row**). We show the training environments that generate the batch data on the left. On the right, we show the test environments where the agents perform behaviors that require the learned policies to leave the data support. In `halfcheetah-jump`, the agent is asked to run while jumping as high as possible given an training offline dataset of halfcheetah running. In `ant-angle`, the ant is rewarded for running forward in a 30 degree angle and the corresponding training offline dataset contains data of the ant running forward directly.

| Environment | Batch Mean | Batch Max | MOPO (ours) | MBPO | SAC | BEAR | BRAC-p | BRAC-v |
|---|---|---|---|---|---|---|---|---|
| halfcheetah-jump | -1022.6 | 1808.6 | **4016.6±144** | 2971.4±1262 | -3588.2±1436 | 16.8±60 | 1069.9±232 | 871±41 |
| ant-angle | 866.7 | 2311.9 | **2530.9±137** | 13.6±66 | -966.4±778 | 1658.2±16 | 1806.7±265 | 2333±139 |

Table 2: Average returns `halfcheetah-jump` and `ant-angle` that require out-of-distribution policy. The MOPO results are averaged over 6 random seeds, $\pm$ standard deviation, while the results of other methods are averaged over 3 random seeds. We include the mean and max undiscounted return of the episodes in the batch data (under Batch Mean and Batch Max, respectively) for comparison. Note that Batch Mean and Max are significantly lower than on-policy SAC, suggesting that the behaviors stored in the buffers are far from optimal and the agent needs to go beyond the data support in order to achieve better performance. As shown in the results, MOPO outperforms all the baselines by a large margin, indicating that MOPO is effective in generalizing to out-of-distribution states where model-free offline RL methods struggle.

## 6 Conclusion

In this paper, we studied model-based offline RL algorithms. We started with the observation that, in the offline setting, existing model-based methods significantly outperform vanilla model-free methods, suggesting that model-based methods are more resilient to the overestimation and overfitting issues that plague off-policy model-free RL algorithms. This phenomenon implies that model-based RL has the ability to generalize to states outside of the data support and such generalization is conducive for offline RL. However, online and offline algorithms must act differently when handling out-of-distribution states. Model error on out-of-distribution states that often drives exploration and corrective feedback in the online setting [37] can be detrimental when interaction is not allowed. Using theoretical principles, we develop an algorithm, model-based offline policy optimization (MOPO), which maximizes the policy on a MDP that penalizes states with high model uncertainty. MOPO trades off the risk of making mistakes and the benefit of diverse exploration from escaping the behavioral distribution. In our experiments, MOPO outperforms state-of-the-art offline RL methods in both standard benchmarks [18] and out-of-distribution generalization environments.

Our work opens up a number of questions and directions for future work. First, an interesting avenue for future research to incorporate the policy regularization ideas of BEAR and BRAC into the reward penalty framework to improve the performance of MOPO on narrow data distributions (such as the "medium" datasets in D4RL). Second, it's an interesting theoretical question to understand why model-based methods appear to be much better suited to the batch setting than model-free methods. Multiple potential factors include a greater supervision from the states (instead of only the reward), more stable and less noisy supervised gradient updates, or ease of uncertainty estimation. Our work suggests that uncertainty estimation plays an important role, particularly in settings that demand generalization. However, uncertainty estimation does not explain the entire difference nor does it explain why model-free methods cannot also enjoy the benefits of uncertainty estimation. For those domains where learning a model may be very difficult due to complex dynamics, developing better model-free offline RL methods may be desirable or imperative. Hence, it is crucial to conduct future research on investigating how to bring model-free offline RL methods up to the level of the performance of model-based methods, which would require further understanding where the generalization benefits come from.

## Broader Impact

MOPO achieves significant strides in offline reinforcement learning, a problem setting that is particularly scalable to real-world settings. Offline reinforcement learning has a number of potential application domains, including autonomous driving, healthcare, robotics, and is notably amenable to safety-critical settings where online data collection is costly. For example, in autonomous driving, online interaction with the environment runs the risk of crashing and hurting people; offline RL methods can significantly reduce that risk by learning from a pre-recorded driving dataset collected by a safe behavioral policy. Moreover, our work opens up the possibility of learning policies offline for new tasks for which we do not already have expert data.

However, there are still risks associated with applying learned policies to high-risk domains. We have shown the benefits of explicitly accounting for error, but without reliable out-of-distribution uncertainty estimation techniques, there is a possibility that the policy will behave unpredictably when given a scenario it has not encountered. There is also the challenge of reward design: although the reward function will typically be under the engineer's control, it can be difficult to specify a reward function that elicits the desired behavior and is aligned with human objectives. Additionally, parametric models are known to be susceptible to adversarial attacks, and bad actors can potentially exploit this vulnerability. Advances in uncertainty quantification, human-computer interaction, and robustness will improve our ability to apply learning-based methods in safety-critical domains.

Supposing we succeed at producing safe and reliable policies, there is still possibility of negative societal impact. An increased ability to automate decision-making processes may reduce companies' demand for employees in certain industries (e.g. manufacturing and logistics), thereby affecting job availability. However, historically, advances in technology have also created new jobs that did not previously exist (e.g. software engineering), and it is unclear if the net impact on jobs will be positive or negative.

Despite the aforementioned risks and challenges, we believe that offline RL is a promising setting with enormous potential for automating and improving sequential decision-making in highly impactful domains. Currently, much additional work is needed to make offline RL sufficiently robust to be applied in safety-critical settings. We encourage the research community to pursue further study in uncertainty estimation, particularly considering the complications that arise in sequential decision problems.

## Acknowledgments and Disclosure of Funding

We thank Michael Janner for help with MBPO and Aviral Kumar for setting up BEAR and D4RL. TY is partially supported by Intel Corporation. CF is a CIFAR Fellow in the Learning in Machines and Brains program. TM and GT are also partially supported by Lam Research, Google Faculty Award, SDSI, and SAIL.

## Footnotes

[2]The definition here extends the definition of admissible confidence interval in [63] slightly to the setting of stochastic dynamics.

[3]If the reward function is known, we do not have to estimate the reward. The theory in Sections 4.1 and 4.2 applies to the case where the reward function is known. To extend the theory to an unknown reward function, we can consider the reward as being concatenated onto the state, so that the admissible error estimator bounds the error on $(s', r)$, rather than just $s'$.

[4]Designing prediction confidence intervals with strong theoretical guarantees is challenging and beyond the scope of this work, which focuses on using uncertainty quantification properly in offline RL.

[5]Code is released at https://github.com/tianheyu927/mopo.

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
