[Supplementary Material]

# Appendix

## A   Reminders about integral probability metrics

Let $(\mathcal{X}, \Sigma)$ be a measurable space. The integral probability metric associated with a class $\mathcal{F}$ of (measurable) real-valued functions on $\mathcal{X}$ is defined as

$$d_{\mathcal{F}}(P, Q) = \sup_{f \in \mathcal{F}} \left| \int_{\mathcal{X}} f \, dP - \int_{\mathcal{X}} f \, dQ \right| = \sup_{f \in \mathcal{F}} \left| \mathop{\mathbb{E}}_{X \sim P}[f(X)] - \mathop{\mathbb{E}}_{Y \sim Q}[f(Y)] \right|$$

where $P$ and $Q$ are probability measures on $\mathcal{X}$. We note the following special cases:

(i) If $\mathcal{F} = \{f : \|f\|_{\infty} \leq 1\}$, then $d_{\mathcal{F}}$ is the *total variation distance*
$$d_{\mathcal{F}}(P, Q) = D_{\mathrm{TV}}(P, Q) := \sup_{A \in \Sigma} |P(A) - Q(A)|$$

(ii) If $\mathcal{F}$ is the set of 1-Lipschitz function w.r.t. to some cost function (metric) $c$ on $\mathcal{X}$, then $d_{\mathcal{F}}$ is the *1-Wasserstein distance* w.r.t. the same metric:
$$d_{\mathcal{F}}(P, Q) = W_1(P, Q) := \inf_{\gamma \in \Gamma(P, Q)} \int_{\mathcal{X}^2} c(x, y) \, d\gamma(x, y)$$

where $\Gamma(P, Q)$ denotes the set of all *couplings* of $P$ and $Q$, i.e. joint distributions on $\mathcal{X}^2$ which have marginals $P$ and $Q$.

(iii) If $\mathcal{F} = \{f : \|f\|_{\mathcal{H}} \leq 1\}$ where $\mathcal{H}$ is a reproducing kernel Hilbert space with kernel $k$, then $d_{\mathcal{F}}$ is the *maximum mean discrepancy*:
$$d_{\mathcal{F}}(P, Q) = \mathrm{MMD}(P, Q) := \sqrt{\mathbb{E}[k(X, X')] - 2\mathbb{E}[k(X, Y)] + \mathbb{E}[k(Y, Y')]}$$

where $X, X' \sim P$ and $Y, Y' \sim Q$.

In the context of Section 4.1, we have (at least) the following instantiations of Assumption 4.2:

(i) Assume the reward is bounded by $r_{\max}$. Then (since $\|V_M^{\pi}\|_{\infty} \leq \frac{r_{\max}}{1-\gamma}$)
$$|G_{\widehat{M}}^{\pi}(s, a)| \leq \frac{r_{\max}}{1 - \gamma} D_{\mathrm{TV}}(\widehat{T}(s, a), T(s, a))$$

This corresponds to $c = \frac{r_{\max}}{1-\gamma}$ and $\mathcal{F} = \{f : \|f\|_{\infty} \leq 1\}$.

(ii) Assume $V_M^{\pi}$ is $L_v$-Lipschitz. Then
$$|G_{\widehat{M}}^{\pi}(s, a)| \leq L_v W_1(\widehat{T}(s, a), T(s, a))$$

This corresponds to $c = L_v$ and $\mathcal{F} = \{f : f \text{ is 1-Lipschitz}\}$.

(iii) Assume $\|V_M^{\pi}\|_{\mathcal{H}} \leq \nu$. Then
$$|G_{\widehat{M}}^{\pi}(s, a)| \leq \nu \mathrm{MMD}(\widehat{T}(s, a), T(s, a))$$

This corresponds to $c = \nu$ and $\mathcal{F} = \{f : \|f\|_{\mathcal{H}} \leq 1\}$.

## B   Proofs

We provide a proof for Lemma 4.1 for completeness. The proof is essentially the same as that for [44, Lemma 4.3].

*Proof.* Let $W_j$ be the expected return when executing $\pi$ on $\widehat{T}$ for the first $j$ steps, then switching to $T$ for the remainder. That is,

$$W_j = \mathop{\mathbb{E}}_{\substack{a_t \sim \pi(s_t) \\ t < j: s_{t+1} \sim \widehat{T}(s_t, a_t) \\ t \geq j: s_{t+1} \sim T(s_t, a_t)}} \left[ \sum_{t=0}^{\infty} \gamma^t r(s_t, a_t) \right]$$

Note that $W_0 = \eta_M(\pi)$ and $W_\infty = \eta_{\widehat{M}}(\pi)$, so

$$\eta_{\widehat{M}}(\pi) - \eta_M(\pi) = \sum_{j=0}^{\infty}(W_{j+1} - W_j)$$

Write

$$W_j = R_j + \mathop{\mathbb{E}}_{s_j,a_j \sim \pi,\widehat{T}}\left[\mathop{\mathbb{E}}_{s_{j+1}\sim T(s_t,a_t)}[\gamma^{j+1}V_M^\pi(s_{j+1})]\right]$$

$$W_{j+1} = R_j + \mathop{\mathbb{E}}_{s_j,a_j \sim \pi,\widehat{T}}\left[\mathop{\mathbb{E}}_{s_{j+1}\sim \widehat{T}(s_t,a_t)}[\gamma^{j+1}V_M^\pi(s_{j+1})]\right]$$

where $R_j$ is the expected return of the first $j$ time steps, which are taken with respect to $\widehat{T}$. Then

$$W_{j+1} - W_j = \gamma^{j+1}\mathop{\mathbb{E}}_{s_j,a_j\sim\pi,\widehat{T}}\left[\mathop{\mathbb{E}}_{s'\sim\widehat{T}(s_j,a_j)}[V_M^\pi(s')] - \mathop{\mathbb{E}}_{s'\sim T(s_j,a_j)}[V_M^\pi(s')]\right]$$

$$= \gamma^{j+1}\mathop{\mathbb{E}}_{s_j,a_j\sim\pi,\widehat{T}}\left[G_{\widehat{M}}^\pi(s_j,a_j)\right]$$

Thus

$$\eta_{\widehat{M}}(\pi) - \eta_M(\pi) = \sum_{j=0}^{\infty}(W_{j+1} - W_j)$$

$$= \sum_{j=0}^{\infty}\gamma^{j+1}\mathop{\mathbb{E}}_{s_j,a_j\sim\pi,\widehat{T}}\left[G_{\widehat{M}}^\pi(s_j,a_j)\right]$$

$$= \gamma\mathop{\bar{\mathbb{E}}}_{(s,a)\sim\rho_{\widehat{T}}^\pi}\left[G_{\widehat{M}}^\pi(s,a)\right]$$

as claimed. □

Now we prove Theorem 4.2.

*Proof.* We first note that a two-sided bound follows from Lemma 4.1:

$$|\eta_{\widehat{M}}(\pi) - \eta_M(\pi)| \leq \gamma\mathop{\bar{\mathbb{E}}}_{(s,a)\sim\rho_{\widehat{T}}^\pi}|G_{\widehat{M}}^\pi(s,a)| \leq \lambda\mathop{\bar{\mathbb{E}}}_{(s,a)\sim\rho_{\widehat{T}}^\pi}[u(s,a)] = \lambda\epsilon_u(\pi) \qquad (12)$$

Then we have, for any policy $\pi$,

$$\eta_M(\hat\pi) \geq \eta_{\widetilde{M}}(\hat\pi) \qquad\qquad\qquad (\text{by } (7))$$
$$\geq \eta_{\widetilde{M}}(\pi) \qquad\qquad\qquad (\text{by definition of } \hat\pi)$$
$$= \eta_{\widehat{M}}(\pi) - \lambda\epsilon_u(\pi)$$
$$\geq \eta_M(\pi) - 2\lambda\epsilon_u(\pi) \qquad\qquad\qquad (\text{by } (12))$$

□

## C   MOPO Practical Algorithm Outline

We outline the practical MOPO algorithm in Algorithm 2.

## D   Ablation Study

To answer question (3), we conduct a thorough ablation study on MOPO. The main goal of the ablation study is to understand how the choice of reward penalty affects performance. We denote **no ens.** as a method without model ensembles, **ens. pen.** as a method that uses model ensemble disagreement as the reward penalty, **no pen.** as a method without reward penalty, and **true pen.** as

---
**Algorithm 2** MOPO instantiation with regularized probabilistic dynamics and ensemble uncertainty
---
**Require:** reward penalty coefficient $\lambda$ rollout horizon $h$, rollout batch size $b$.
1: Train on batch data $\mathcal{D}_{\text{env}}$ an ensemble of $N$ probabilistic dynamics $\{\widehat{T}^i(s', r \,|\, s, a) = \mathcal{N}(\mu^i(s, a), \Sigma^i(s, a))\}_{i=1}^N$.
2: Initialize policy $\pi$ and empty replay buffer $\mathcal{D}_{\text{model}} \leftarrow \varnothing$.
3: **for** epoch $1, 2, \dots$ **do** $\qquad\qquad\qquad\qquad\triangleright$ This for-loop is essentially one outer iteration of MBPO
4: $\quad$ **for** $1, 2, \dots, b$ (in parallel) **do**
5: $\quad\quad$ Sample state $s_1$ from $\mathcal{D}_{\text{env}}$ for the initialization of the rollout.
6: $\quad\quad$ **for** $j = 1, 2, \dots, h$ **do**
7: $\quad\quad\quad$ Sample an action $a_j \sim \pi(s_j)$.
8: $\quad\quad\quad$ Randomly pick dynamics $\widehat{T}$ from $\{\widehat{T}^i\}_{i=1}^N$ and sample $s_{j+1}, r_j \sim \widehat{T}(s_j, a_j)$.
9: $\quad\quad\quad$ Compute $\tilde{r}_j = r_j - \lambda \max_{i=1}^N \|\Sigma^i(s_j, a_j)\|_{\text{F}}$.
10: $\quad\quad\quad$ Add sample $(s_j, a_j, \tilde{r}_j, s_{j+1})$ to $\mathcal{D}_{\text{model}}$.
11: $\quad$ Drawing samples from $\mathcal{D}_{\text{env}} \cup \mathcal{D}_{\text{model}}$, use SAC to update $\pi$.
---

a method using the true model prediction error $\|\widehat{T}(s, a) - T(s, a)\|$ as the reward penalty. Note that we include **true pen.** to indicate the upper bound of our approach. Also, note that **no ens.** measures disagreement among the ensemble: precisely, if the models' mean predictions are denoted $\mu_1, \dots, \mu_N$, we compute the average $\bar{\mu} = 1/N \sum_{i=1}^N \mu_i$ and then take $\max_i \|\mu_i - \bar{\mu}\|$ as the ensemble penalty.

The results of our study are shown in Table 3. For different reward penalty types, reward penalties based on learned variance perform comparably to those based on ensemble disagreement in D4RL environments while outperforming those based on ensemble disagreement in out-of-distribution domains. Both reward penalties achieve significantly better performances than no reward penalty, indicating that it is imperative to consider model uncertainty in batch model-based RL. Methods that uses oracle uncertainty obtain slightly better performance than most of our methods. Note that **MOPO** even attains the best results on halfcheetah-jump. Such results suggest that our uncertainty quantification on states is empirically successful, since there is only a small gap. We believe future work on improving uncertainty estimation may be able to bridge this gap further. Note that we do not report the results of methods with oracle uncertainty on walker2d-mixed and ant-angle as we are not able to get the true model error from the simulator based on the pre-recorded dataset.

In general, we find that performance differences are much larger for halfcheetah-jump and ant-angle than the D4RL halfcheetah-mixed and walker2d-mixed datasets, likely because halfcheetah-jump and ant-angle requires greater generalization and hence places more demands on the accuracy of the model and uncertainty estimate.

Finally, we perform another ablation study on the choice of the reward penalty. We consider the $u^{\text{mean}}(s, a) = \frac{1}{N} \sum_{i=1}^N \|\Sigma_\phi^i(s, a)\|_{\text{F}}$, the average standard deviation of the learned models in the ensemble, as the reward penalty instead of the max standard deviation as used in MOPO. We denote the variant of MOPO with the average learned standard deviation as **MOPO, avg. var.**. We compare **MOPO** to **MOPO, avg. var.** in the halfcheetah-jump domain. **MOPO** achieves **4140.6**±88 average return while **MOPO, avg. var.** achieves **4166.3**±228.8 where the results are averaged over 3 random seeds. The two methods did similarly, suggesting that using either mean variance or max variance would be a reasonable choice for penalizing uncertainty.

# E  Empirical results on generalization capabilities

We conduct experiments in *ant-angle* to show the limit of MOPO's generalization capabilties. As shown in Table 4, we show that MOPO generalizes to Ant running at a $45°$ angle (achieving almost buffer max score), beyond the $30°$ shown in the paper, while failing to generalize to a $60$ and $90°$ degree angle. This suggests that if the new task requires to explore states that are completely out of the data support, i.e. the buffer max and buffer mean both fairly bad, MOPO is unable to generalize.

| Method | halfcheetah-mixed | walker2d-mixed | halfcheetah-jump | ant-angle |
|---|---|---|---|---|
| **MOPO** | $6405.8 \pm 35$ | $1916.4 \pm 611$ | $4016.6 \pm 144$ | $2530.9 \pm 137$ |
| **MOPO, ens. pen.** | $6448.7 \pm 115$ | $1923.6 \pm 752$ | $3577.3 \pm 461$ | $2256.0 \pm 288$ |
| **MOPO, no pen.** | $6409.1 \pm 429$ | $1421.2 \pm 359$ | $-980.8 \pm 5625$ | $18.6 \pm 49$ |
| **MBPO** | $5598.4 \pm 1285$ | $1021.8 \pm 586$ | $2971.4 \pm 1262$ | $13.6 \pm 65$ |
| **MBPO, no ens.** | $2247.2 \pm 581$ | $500.3 \pm 34$ | $-68.7 \pm 1936$ | $-720.1 \pm 728$ |
| **MOPO, true pen.** | $6984.0 \pm 148$ | N/A | $3818.6 \pm 136$ | N/A |

Table 3: Ablation study on two D4RL tasks `halfcheetah-mixed` and `walker2d-mixed` and two out-of-distribution tasks `halfcheetah-jump` and `ant-angle`. We use average returns where the results of MOPO and its variants are averaged over 6 random seeds and MBPO results are averaged over 3 random seeds as in Table 2. We observe that different reward penalties can all lead to substantial improvement of the performance and reward penalty based on learned variance is a better choice than that based on ensemble disagreement in out-of-distribution cases. Methods that use oracle uncertainty as the reward penalty achieve marginally better performance than **MOPO**, implying that **MOPO** is effective at estimating the uncertainty.

| Environment | Buffer Max | Buffer Mean | MOPO |
|---|---|---|---|
| `ant-angle-45` | 3168.7 | 1105.5 | $2571.3 \pm 598.1$ |
| `ant-angle-60` | 1953.7 | 846.7 | $840.5 \pm 1103.7$ |
| `ant-angle-90` | 838.8 | -901.6 | $-503.2 \pm 803.4$ |

Table 4: Limit of generalization on `ant-angle`.

# F    Experiments on HIV domains

Beyond continous control tasks in MuJoCo, we test MOPO on an HIV treatment simulator slightly modified from the one in the whynot package. The task simulates the sequential decision making in HIV treatment, which involves determining the amounts of two anti-HIV drugs to be administered to the patient in order to maximize the immune response and minimize the amount of virus. The agent observes both of those quantities as well as the (log) number of infected and uninfected T cells and macrophages.

We evaluated MOPO with the data generated from the first 200k steps of training an online SAC agent on this environment. We show results in Table 5, where MOPO outperforms BEAR and achieves almost the buffer max score.

| Buffer Max | Buffer Mean | SAC (online) | BEAR | MOPO |
|---|---|---|---|---|
| 15986.2 | 6747.2 | $25716.3 \pm 254.3$ | $11709.1 \pm 1292.1$ | $\mathbf{13484.6} \pm 3900.7$ |

Table 5: HIV treatment results, averaged over 3 random seeds.

# G    Experiment Details

## G.1    Details of out-of-distribution environments

For `halfcheetah-jump`, the reward function that we use to train the behavioral policy is $r(s, a) = \max\{v_x, 3\} - 0.1 * \|a\|_2^2$ where $v_x$ denotes the velocity along the x-axis. After collecting the offline dataset, we relabel the reward function to $r(s, a) = \max\{v_x, 3\} - 0.1 * \|a\|_2^2 + 15 * (z - \text{init z})$ where $z$ denotes the z-position of the half-cheetah and init z denotes the initial z-position.

For `ant-angle`, the reward function that we use to train the behavioral policy is $r(s, a) = v_x - $ control cost. After collecting the offline dataset, we relabel the reward function to $r(s, a) = v_x \cdot \cos\frac{\pi}{6} + v_y \cdot \sin\frac{\pi}{6} - $ control cost where $v_x, v_y$ denote the velocity along the $x, y$-axis respectively.

For both out-of-distribution environments, instead of sampling actions from the learned policy during the model rollout (line 10 in Algorithm 2), we sample random actions from $\text{Unif}[-1, 1]$, which achieves better performance empirically. One potential reason is that using random actions during model rollouts leads to better exploration of the OOD states.

| Dataset type | Environment | MOPO $(h, \lambda)$ | MBPO $h$ |
|---|---|---|---|
| random | halfcheetah | 5, 0.5 | 5 |
| random | hopper | 5, 1 | 5 |
| random | walker2d | 1, 1 | 5 |
| medium | halfcheetah | 1, 1 | 5 |
| medium | hopper | 5, 5 | 5 |
| medium | walker2d | 5, 5 | 5 |
| mixed | halfcheetah | 5, 1 | 5 |
| mixed | hopper | 5, 1 | 5 |
| mixed | walker2d | 1, 1 | 1 |
| med-expert | halfcheetah | 5, 1 | 5 |
| med-expert | hopper | 5, 1 | 5 |
| med-expert | walker2d | 1, 2 | 1 |

Table 6: Hyperparameters used in the D4RL datasets.

## G.2 Hyperparameters

Here we list the hyperparameters used in the experiments.

For the D4RL datasets, the rollout length $h$ and penalty coefficient $\lambda$ are given in Table 6. We search over $(h, \lambda) \in \{1, 5\}^2$ and report the best final performance, averaged over 3 seeds. The only exceptions are halfcheetah-random and walker2d-medium-expert, where other penalty coefficients were found to work better.

For the out-of-generalization tasks, we use rollout length $5$ for `halfcheetah-jump` and $25$ for `ant-angle`, and penalty coefficient $1$ for `halfcheetah-jump` and $2$ for `ant-angle`.

Across all domains, we train an ensemble of 7 models and pick the best 5 models based on their prediction error on a hold-out set of 1000 transitions in the offline dataset. Each of the model in the ensemble is parametrized as a 4-layer feedforward neural network with 200 hidden units and after the last hidden layer, the model outputs the mean and variance using a two-head architecture. Spectral normalization [45] is applied to all layers except the head that outputs the model variance.

For the SAC updates, we sample a batch of 256 transitions, $5\%$ of them from $\mathcal{D}_{\text{env}}$ and the rest of them from $\mathcal{D}_{\text{model}}$. We also perform ablation studies on the percentage of the real data in a batch for MOPO. For simplicity, we use MBPO, which essentially MOPO without reward penalty, for this ablation study. We tried to train MBPO with data all sampled from $\mathcal{D}_{\text{model}}$ and no data from $\mathcal{D}_{\text{env}}$ and compare the performance to MBPO with $5\%$ of data from $\mathcal{D}_{\text{env}}$ on all 12 settings in the D4RL benchmark. We find that the performances of both methods are not significantly distinct: no-real-data MBPO outperforms $5\%$-real-data MBPO on 6 out of 12 tasks and lies within one SD of $5\%$-real-data MBPO on 9 out of 12 tasks.