[Reviews · NeurIPS 2020]

Review 1

Summary and Contributions: The authors present MOPO, a model-based method which uses an ensemble of models to provide an uncertainty estimate over the next-state distribution. This scheme balances between return maximization and avoidance of regions lacking data. They obtain sota performance across most domains with feature-based states.

Strengths: - Theoretical analysis of the penalized policy, showing that it is better than ignoring the penality and exploiting the models errors. - Multiple experiments against multiple baselines and domains.

Weaknesses: The main weakness in my opinion is that this work is fairly limited in terms of applicability. In order to provide an uncertainty estimate they need to provide an analytical representation of the next-state distribution. In this work they assume it is Gaussian. In certain settings this may be correct, but in the general case, and especially when external disturbances exist, this assumption may no longer hold. This limits the types of models that can be used. For instance, the ability to predict a distribution over the next-states may prove difficult when the feature representation isn't provided, but rather one must learn directly from raw-inputs (e.g., images).

Correctness: The claims seem correct and the empirical methodology looks ok.

Clarity: Yes

Relation to Prior Work: Yes

Reproducibility: Yes

Additional Feedback: As stated above, the main issue in my opinion is in the ability to extend this work to more general settings. I know that the research community likes to focus on mujoco-like tasks, however, their simplicity isn't always a good sign. I think the addition of the theoretical analysis is nice. However, it would surprise me if previous works haven't suggest similar methods. The idea of avoiding regions with high uncertainty seems complementary to previous works in RL, in the online setting, where the agent is pushed towards collecting data in these areas. --- POST REBUTTAL --- I would like to thank the authors for the additional experiments. Not trivial given the timeframe. I think this paper is in the right direction, but is currently weak in the sense that there is a large gap between theory and practice. In addition, as I argued in the original review - this method currently works well only since it is EASY to quantify the disagreement between models in physical domains (e.g., the L2 distance is a good approximation of the underlying metric). When the model is tasked to predict the next IMAGE it isn't clear what is the proper way to quantify disagreement and especially when considering stochasticity.


Review 2

Summary and Contributions: The paper presents a model-based offline reinforcement learning algorithm that builds on MBPO. It constructs a lower bound on the return in the true MDP based on a general integral probability metric between the transition probabilities of the true and learned MDP. Motivated by this result, the paper proposes a practical method to optimize a uncertainty-penalized MDP.

Strengths: The paper tackles offline reinforcement learning, a problem that is relevant to the NeurIPS community. It provides a theoretically sound motivation for how model errors can affect the estimated reward function and proposes a novel solution to the problem. Moreover, it evaluates the problem on an interesting out-of-distribution task that likely pushes on the limits of what model-generalization can achieve.

Weaknesses: As is common with theorems in the style of L 4.3, the bound is very loose in general. In particular Lambda >> r_max in (3). This is largely owed to the generality of assumption 4.1, which includes a maximum over all possible value functions. A more fine-grained analysis that incorporates the effect that model errors have on the difference in value function would likely lead to more interesting results. The practical algorithm is fairly disconnected from the theoretical motivation. In particular, epistemic uncertainty is arguably the most important factor here, since that is what quantifies whether we have seen sufficient data. In contrast, the aleatoric uncertainty that the practical implementation in Sec. 4.3 considers is something that is averaged over in our performance metric and thus of lesser concern. More importantly, for the deterministic Mujoco environments that the paper evaluates, the aleatoric uncertainty Sigma_psi^I should generally be zero (though likely will not be for the learned model). That MOPO still has higher performance over MBPO likely hints at that either something other than the theoretical motivation is helping performance or that the benchmarks are not interesting to evaluate offline RL properly. In the appendix there are some evaluations on using the true penalty and an ensemble disagreement, but little information is provided on what these are precisely.

Correctness: The theoretical claims are correct.

Clarity: The paper is well-written and easy to follow. The only caveat to this is that, at several points in the introduction and related work, the paper throws up to 7 references at once at the reader. This is poor style and should be avoided.

Relation to Prior Work: The relation to prior work is discussed.

Reproducibility: Yes

Additional Feedback: Concrete questions for rebuttal: Could you explain how "true pen." and "ensemble pen" in the appendix were computed for the experiments? How did you apply MBPO to the problem? Did you always take the full offline dataset as the "environment data" for MBPO? In that case is it fair to say that the only thing that forces MBPO to stay close to the offline data is the expectation over the reward from the ensemble model? Do you have any intuition for how far the model generalizes? E.g., in the ant environment are 30 degrees the limit or does the model also generalize to e.g., walking backwards? Suggestions for improvement: Make the contribution of aleatoric v.s. empistemic uncertainty in the learning process clear and evaluate how these difference uncertainties contribute to generalization (or lack thereof). It would be nice to compare against something like PETs in [7] that relies only on model-rollouts to optimize the policy, rather than also throwing the offline data into the optimization. ----------- post-rebuttal Thank you for the thorough answers. It would be nice to use the extra page provided upon acceptance to include the additional discussion. The main weakness of the paper in my eyes remains the weak link between theory and experiments.


Review 3

Summary and Contributions: Authors extend model-based policy optimisation (MBPO) by arguing that using a policy from a penalised version of the learned MDP is favourable, if the penalty reflects the risk of using a poor dynamics predictor. Authors develop some theoretical results on the stability of the resulting framework, that rely on the ability to restrict the space of value functions (or/and their generating policy space) and the existence of a bound on the effects of the transition function impression. Failing to put such restrictions to practice, authors suggest a penalty function that they later show to work well in practice, at the cost of losing theoretical properties. Experimental data indeed shows that the suggested framework with the empirical penalty (MOPO) is indeed effective in direct performance measurement and a form of transfer-learning.

Strengths: Theoretical aspects of the paper appear to be sound and correct. The initial promise of the suggested framework and its generality were very enticing, and the results began to promise theoretical stability and performance guarantees that can hardly be found elsewhere. The practical form of the approach, though carrying only basic structural similarity to the theoretical counterpart, is shown to be surprisingly effective. Experiments also suggest delineating application areas of off-line and on-line RL approaches -- a nice bonus.

Weaknesses: The framework follows a popular trend of these days of attaching an additional penalty to the system, based on various forms and representations of performance uncertainty information. The fact that it is applied here to off-line RL makes it no fresher an idea. The vast chasm between the theory and the actual MOPO underlines this weakness. Without theoretical backing, MOPO clearly becomes "we tried this regularisation and it worked" -- of which there are many examples. Theory was supposed to save the day by saying "we tried this _because_...., and that's why it works". Alas, that obviously did not happen.

Correctness: Claims are well formulated, and their proofs appear to be correct. Empirical methodology is well through through, though it is detached from the theoretical contribution, and should be taken on its own.

Clarity: The paper is very well written. Though much detail needed for reproduction of this research will appear only in the Appendix, the paper is readable on its own. It is a pity thought, that the explicit description of the paper's actual major contribution -- the practical MOPO -- has to be delegated to the appendix as well. The writing is well staged, inspires further investigation of the subject and suggests potential links to other areas.

Relation to Prior Work: The work is clearly and well positioned.

Reproducibility: Yes

Additional Feedback: ----- post-rebuttal ---- If "argue that ... justifies", then please do so explicitly. The rest of your response suggests that your practical penalty surpasses your theory far beyond its current application. A great promise, but it only serves to underline again how detached your theory is from your practice. Kudos on the practical intuition that goes beyond your ability to explain it.


Review 4

Summary and Contributions: The authors study model based RL in the context of offline RL. Since model based methods are trained with supervised loss which could provide a better learning signal, as compared to using RL. Another benefit of using model based RL for offline RL is that it could provide better out of distribution generalization to new states and actions, as compared to model free RL, where most of the recent methods constraint the policy to the data distribution and hence can't generalize to out of distribution states. The proposed method extends MOPO and uses the max variance of the lipschitz constrained model as a penalty to the reward function (as compared to mean). Empirically the paper shows: - Model based methods can perform well for offline RL as compared to model-free methods. Something which seems reasonable as using model based methods for offline RL is like a "supervised" learning problem. - The proposed method can generalize better to out of distribution states and actions.

Strengths: Studies the problem of offline RL using ideas from model based RL. The proposed algorithm has some theoretical justification that it maximizes expected return corresponding to a MDP that penalizes states with high model uncertainty.

Weaknesses: The authors write "while this estimator lacks theoretical guarantees, we find that it is sufficiently accurate to achieve good performance" I'm curious what happens if authors choose 1) mean variance as compared to max variance for penalizing the reward 2) disagreement b/w different model predictions on a particular data point (without enforcing the lipschitz constraint.

Correctness: The method seems to be correct

Clarity: The paper is well written and easy to follow.

Relation to Prior Work: Prior work has been well cited and discussed.

Reproducibility: Yes

Additional Feedback:

[Author Response · NeurIPS 2020]

We thank all the reviewers for the constructive feedback. We will incorporate the valuable suggestions in the revised
version. Below, we respond to all of the reviewer comments, including multiple new experiments as requested:

**R1:** *"fairly limited in terms of applicability... the ability to extend this work to more general settings?"* Since
submission, we have tested MOPO on a non-MuJoCo environment: an HIV treatment simulator slightly modified
from the one in the whynot package. The task simulates the sequential decision making in HIV treatment. We
evaluated MOPO with the data generated from the first 200k steps of training an online SAC agent on this envi-
ronment. We show results in Table 1, where MOPO outperforms BEAR and achieves almost the buffer max score.

While the particular choice of $u(s,a)$
that we used in our experiments makes
use of the Gaussianity of the dynamics
model, this is not a fundamental require-

| Buffer Max | Buffer Mean | SAC (online) | BEAR | MOPO |
|---|---|---|---|---|
| 15986.2 | 6747.2 | $25716.3 \pm 254.3$ | $11709.1 \pm 1292.1$ | $\mathbf{13484.6 \pm 3900.7}$ |

Table 1: HIV treatment results, averaged over 3 random seeds.

ment – one could eschew Gaussian models and estimate model uncertainty some other way, such as model ensemble
disagreement (which we tried; see Appendix E).

**R4:** *"Try 1) mean variance as compared to max variance for penalizing the reward or 2) disagreement b/w different*
*model predictions"* 1) We added comparison between max variance and mean variance as the reward penalty in the
halfcheetah-jump domain. MOPO with max variance achieves $\mathbf{4140.6} \pm 88$ average return while MOPO with mean
variance achieves $\mathbf{4166.3} \pm 228.8$. The two methods did similarly, suggesting that using either mean variance or max
variance would be a reasonable choice for penalizing uncertainty. 2) Table 3 in Appendix E of the paper show the
results of using model ensemble disagreement without Lipschitz regularization (denoted as MOPO, no Lip, ens. Pen.).
It performs similarly to MOPO in D4RL experiments but worse than MOPO on out-of-distribution generalization tasks.

**R2:** *"intuition for how far the model generalizes?"* We added experiments in Table 2 that show that MOPO generalizes
to Ant running at a $45°$ angle (achieving almost buffer max score), beyond the $30°$ shown in the paper, while failing to
generalize to a 60 and $90°$ degree angle. This suggests that if the new task requires to explore states that are completely
out of the data support, i.e. the buffer max and buffer mean both fairly bad, MOPO is unable to generalize.

**R2:** *"How were 'true pen.' and 'ensemble pen.' in the appendix*
*computed?"* As explained on line 593-595 in Appendix E,
"true pen." is computed as the model prediction error $\|T(s,a) -$
$\widehat{T}(s,a)\|$ using the ground truth dynamics $T$. The "ensemble
pen." measures disagreement among the ensemble: precisely,

| Environment | Buffer Max | Buffer Mean | MOPO |
|---|---|---|---|
| ant-angle-45 | 3168.7 | 1105.5 | $2571.3 \pm 598.1$ |
| ant-angle-60 | 1953.7 | 846.7 | $840.5 \pm 1103.7$ |
| ant-angle-90 | 838.8 | -901.6 | $-503.2 \pm 803.4$ |

Table 2: Limit of generalization on ant-angle.

if the models' mean predictions are denoted $\mu_1, \ldots, \mu_N$, we compute the average $\bar{\mu} = 1/N \sum_{i=1}^{N} \mu_i$ and then take
$\max_i \|\mu_i - \bar{\mu}\|$ as the ensemble penalty. We will make sure these explanations appear prominently in the main paper.

**R2:** *"How did you apply MBPO to the problem?"* As discussed on line 140-149, we first use the full offline dataset
to train the model and then use the trained model for model rollouts to optimize the policy. There is no explicit
regularization that forces MBPO to stay close to the offline data, but maximizing the expectation over the reward of the
trajectories generated from the rollouts of the ensemble model can be viewed as some sort of implicit regularization
since the learned model learns the transition dynamics induced by the offline data.

**R2:** *"It would be nice to compare against something... that relies only on model-rollouts to optimize the policy."* In our
experiments, when sampling from the replay buffers, only a small fraction (5%) comes from the real data, and the rest
from the model-generated data. For further comparison, we re-ran MBPO with only model-generated data on the D4RL
tasks and found that its performance was not significantly affected: no-real-data MBPO outperforms 5%-real-data
MBPO on 6/12 tasks and lies within one SD of 5%-real-data MBPO on 9/12 tasks.

**R2, R3:** *"The practical algorithm is fairly disconnected from the theoretical motivation... The vast chasm between the*
*theory and the actual MOPO?"* We would argue that the theory motivates and justifies the particular way of penalizing
the reward using the uncertainty estimates of the dynamics. Indeed, we didn't provide any theory for the uncertainty
estimate of the dynamics, but provable uncertainty quantification for nonlinear supervised learning is a major and
modular open question in statistics and ML, which is beyond the scope of this paper.

**R2:** *"A more fine-grained analysis that incorporates the effect that model errors have on the difference in value function*
*would likely lead to more interesting results?"* This is true – certainly $R_{\max}/(1-\gamma)$ is a loose bound. The main
difficulty seems to be that without any assumptions on the value function (other than boundedness), the difference could
theoretically be arbitrary if the model has any error. If the value function is Lipschitz, we get a bound that involves the
1-Wasserstein distance, which is more fine-grained than total variation distance in the sense that it incorporates the
magnitude of error according to the geometry of the state space. However, we do not expect the value function to be
Lipschitz in general. A possible strategy would be to use $V_{\widehat{M}}^{\pi}$, which we can approximate using only samples from the
model, to estimate a bound on the difference in $V_M^{\pi}$. We leave this for future work.

[Meta-Review · NeurIPS 2020]

The authors present a model-based methods (MOPO) that uses an ensemble of methods to provide an uncertainty estimate over the next state distribution. The model can be used to avoid uncertain areas in state space, thus allowing a lower bound on the return in the true MDP. The reviewers are unanimously positive about the paper. The reviewers mention the theoretical soundness of the motivation and the novelty of the analysis of the penalised policy, as well as the thoroughness of the empirical evaluation. The reviewers do mention certain limitation, like needing an analytic representation of the next state distribution and the ability to quantify disagreement. The looseness of the bound was also mentioned. But most importantly is the offset between the theoretical results and the practical implementation. Nevertheless, the empirical strength as well as the novelty of the theoretical results were deemed by the reviewers to outweigh the possible limitations, and I’m happy to go along with the reviewers and recommend the paper for acceptance. Several of the reviewer’s questions were answered in the rebuttal. Please update the final version of the paper to include the additional results and clarifications.